

# A novel deep learning model for predicting marine pollution for sustainable ocean management

Michael Onyema Edeh[1], Surjeet Dalal[2], Musaed Alhussein[3], Khursheed Aurangzeb[3], Bijeta Seth[4] and Kuldeep Kumar[5]

[1] Department of Mathematics and Computer Science, Coal City University, Enugu, Nigeria
[2] Department of Computer Science and Engineering, Amity University Haryana, Gurugram, Haryana, India
[3] Department of Computer Engineering, King Saud University, Riyadh, Saudi Arabia
[4] Dept. of Computer Science and Engineering, B.M. Institute of Engineering & Technology, Sonipat, Haryana, India
[5] Solutions Architect AI, Fourien Inc., Alberta, Edmonton, Canada

## ABSTRACT

Climate change has become a major source of concern to the global community. The steady pollution of the environment including our waters is gradually increasing the effects of climate change. The disposal of plastics in the seas alters aquatic life. Marine plastic pollution poses a grave danger to the marine environment and the long-term health of the ocean. Though technology is also seen as one of the contributors to climate change many aspects of it are being applied to combat climate-related disasters and to raise awareness about the need to protect the planet. This study investigated the amount of pollution in marine and undersea leveraging the power of artificial intelligence to identify and categorise marine and undersea plastic wastes. The classification was done using two types of machine learning algorithms: two-step clustering and a fully convolutional network (FCN). The models were trained using Kaggle's plastic location data, which was acquired *in situ*. An experimental test was conducted to validate the accuracy and performance of the trained models and the results were promising when compared to other conventional approaches and models. The model was used to create and test an automated floating plastic detection system in the required timeframe. In both cases, the trained model was able to correctly identify the floating plastic and achieved an accuracy of 98.38%. The technique presented in this study can be a crucial instrument for automatic detection of plastic garbage in the ocean thereby enhancing the war against marine pollution.

## INTRODUCTION

There is a growing concern about climate change and its devastating effects. In recent times, the world has witnessed many disasters which can be traced to climate change. Waters are not spared as they have become a disposal point for all kinds of waste, especially plastic waste (*Espeseth et al., 2020*). This has resulted in serious water pollution (*Bayrakdar, 2020*).

Corresponding author
Surjeet Dalal,
profsurjeetdalal@gmail.com

Ocean pollution has caused a range of environmental problems including organic matter enrichment in confined waters, oil contamination, and sedimentation caused by land-based operations like dredging (*Nunes & Leston, 2020*). Dissolved oxygen (DO) and microbial concentration levels are significant markers of the health of coastal water; however, pollution from organic and inorganic sources has increased over time. Untreated home and industrial waste contributes to Coast pollution by causing bacteria to consume large amounts of oxygen from the saltwater, lowering the concentration of dissolved oxygen in the water and negatively impacting aquatic life. Aquatic lives are now endangered more than before (*Minchew, Jones & Holt, 2012*). Technology offers some possibilities that can assist environmentalists and relevant agencies in dealing with the surfacing problem of plastic waste in water. For instance, artificial intelligence is being deployed for ocean observation and proper monitoring of underwater things and spillage (*Song et al., 2020*). This study is directed towards that regard to automate the process and approach to automatically search and locate undersea plastic wastes. Leveraging the power of artificial intelligence as suggested in the current would go a long way to support the long-term goal of achieving sustainable ocean health both for animals and humans (*Dalal et al., 2024*).

Because of the above ocean pollution and marine debris problems, we raised the following question: can we design a new machine-learning mechanism to predict marine pollution for sustainable ocean health? If we can create such a model then it helps to limit the consequences of microplastics and hazardous pollutants in seafood. There is continuing study throughout the country concentrating on the possible damage to wildlife and humans from trash exposure and ingestion (*Dalal et al., 2023*).

Our contributions can be summarized as follows:

- An optimized Two-step clustering model is proposed to classify the cluster of marine debris correctly.
- The optimization mechanism is used to tune the cluster features (CF) Tree Tuning Criteria for better coastal pollution assessment.
- A modified YOLO model is proposed to detect the underwater trash in the ocean.
- The learning rate hyperparameter tuning mechanism is used to tune the YOLO algorithm to detect the underwater trash in the ocean.
- This proposed algorithm provides a new idea for reducing ocean pollution by properly disposing of plastics and other recyclable materials.

This paper is designed with aiming of predicting marine pollution in two perspectives *i.e.,* coastal pollution and underwater trash. 'Related works' defines the various existing works carried out in this problem domain. The section explains the datasets used for experimental purposes. In this paper, two different types of datasets are being taken. 'Coastal pollution dataset' demonstrated coastal pollution assessment through modified Two-step clustering algorithms and 'Underwater trash dataset' highlighted underwater trash detection with the YOLO algorithm. 'Discussion' demonstrates the results, followed by the conclusion of the paper in 'Conclusion and Future scope'.

# RELATED WORKS

In the beginning, we discussed the promise of using machine learning techniques to address marine debris and pollution. Predictions of underwater pollution and evaluations of coastal pollution have been made using several machine learning algorithms, with positive results.

## Deep learning models for environmental predictions

One of the challenges of machine learning is determining how polluted marine trash is likely to be. Some other contexts are briefly explained below. *Minchew, Jones & Holt (2012)* examined that the dampening of the sea wave terrible portions by the oil and a compelling fall in dielectric steady caused by the mix of 65–90 percent water with oil in the surface layer are to blame for the difference in backscatter over the fundamental smooth. As instrument noise rises over the instrument commotion floor, the anisotropy, A, border exhibits substantial variation throughout the oil spill and a large-reach subordinate sign.

*Gao et al. (2018)* fostered a sea object float forecast model. "High goal surface flows, Stokes float, and winds" were handled, and a progression of model examinations was built. Predicted paths for the items on display were similar to observed directions for the floating objects. Many of the items that were tracked ended up at Reunion Island, Mauritius, and Tanzania with probabilities of 5 percent, 5 percent, and 19 percent, respectively, after drifting north and then west. Eventually, most of the reenactment's components were located in the western Indian Ocean, about latitude 10°S. Possibly owing to the influence of southeast trade winds, there were substantial differences in the results of several room factor explorations.

*Li et al. (2019)* proposed a various levelled structure for coal and gangue recognition because of profound learning models. First, the Gaussian pyramid guideline is utilised to create staggered prepared information, resulting in a variety of scales of coal and gangue image highlights. Next, convolutional neural network (CNNs) are created to recognise coal and gangue objects in discrete up-and-comer regions. Three distinct datasets were used to test our method. Coal and gangue object localization precision improved by 0.8 percent in comparison to previous methods, reaching up to 98.33 percent with the suggested strategy. We also present a methodology that makes it possible to see a large number of coal and gangue items at once and addresses the problem of lining requirements in existing methods.

Stable water maps will soon be available thanks to a recurrence-based technique presented by *Meng et al. (2020)* to distinguish between hydroponics water and conventional water. Each year, Landsat Level-2 images from 1984 to 2018 were used to construct yearly 30 m target water items, which were then used to investigate the spatial-worldly changes in the Taihu Lake area. Furthermore, each big graphical modification was linked to a specific event in the actual world at the time. The outcomes propose that human exercises impact surface water rather than environmental changes in the Taihu Lake locale, and affirm the adequacy of biological security strategy in keeping up with the strength of the aggregate sum of normal water in the beyond couple of years. The spatial-transient unsettling influence of hydroponics likewise gave one more point of view and solid proof of past investigations because of human exercises on the eutrophication interaction of Taihu Lake.

*Melody et al. (2020)* proposed CNN is fit for mining spatial components from a huge informational index. CNN's deep multi-facet component extraction has inspired us to develop an innovative oil slick-detection-proof technique in this research. The PolSAR information is immediately converted into a 9-channel information block for CNN's use. A five-layer CNN is then used to extract two important levels of information from the initial data set. Radial basis function support vector machine (RBF-SVM), a support vector machine approach with an outspread premise work component, is utilised for order. RADARSAT-2 polarimetric SAR data was used in this study to endorse the suggested technique. Findings reveal that the suggested technique has significant effects on grouping accuracy and the kappa coefficient generally. Other advantages include a reduction in deception rates and the ability to distinguish an oil slick from a biogenic smooth.

*Chen et al. (2021)* expressed that diminishing ozone harming substance emanations turns into a first concern on the planet with the development of an unnatural weather change and natural issues. Different sustainable power sources show up during the last many years. Sea catches and stores colossal measures of energy, which could fulfill multiple times of world energy interest. Because of innovation constraints and monetary contemplations, marine ebb and flow energy seems the most alluring decision contrasted and the other sea energy structure. In this paper, it shows the interest and the rule of the marine current energy, and furthermore talks about the benefits and weaknesses. The ecological effects around the gadgets, the mechanical difficulties, and the fundamental help structures are introduced also.

*Xue et al. (2021a)* and *Xue et al. (2021b)* examined whether convolutional neural network can recognize the distinctions of flotsam and jetsam and normal remote ocean climate, to really accomplish remote ocean garbage distinguishing proof. Initial, a genuine remote ocean flotsam and jetsam pictures dataset is developed for additional grouping research in view of a web-based remote ocean trash data set. Moreover, five normal CNN) structures are likewise utilized to execute the order cycle. At long last, the distinguishing proof examinations are done to approve the presentation of the proposed strategy. The outcomes exhibit that the proposed strategy is better than the cutting edge CNN technique and has the potential for remote ocean trash recognizable proof.

*Shi et al. (2021)* developed Wheatstone span circuits to improve the reaction to garbage and lessen the impedance of oil temperature and thickness. The complete examination of the identification brings about two modes can further develop sensor versatility and conquer the lack of low dependability in light of a solitary discovery technique. This sensor can give more precise garbage data to the shortcoming analysis of water driven hardware and is of incredible importance for smart upkeep.

## Marine pollution monitoring and prediction

*Espeseth et al. (2020)* introduced two strategies that are correlative as far as recognizing transient replacements inside an oil spill. In contrasting ways, the two approaches show how various people view oil spills. As an intermediate for increasing oil thickness, the primary technique detects regions within the smooth that demonstrate a determinedly high damping percentage (the distinction between clean ocean and oil power). With each

new photo that is included in the computation, this approach updates the scene's age in addition to the original age. To see this development and its persistence in both broad settings, one must examine the outcomes of these two approaches.

*Walden & Mehrubeoglu (2020)* featured that waterway flotsam and jetsam can affect numerous parts of marine conditions remembering marine routes and sports looking for waterways. Stream plastic flotsam and jetsam contamination are happening at such a huge scope universally that it is grave to follow and measure the sheer amount of lost plastics. Advanced picture handling is one compelling method for observing stream flotsam and jetsam, yet the related complex cycles have intrinsic difficulties. This study examines the effects of evaluating the weight and volume of plastic jug flotsam and jetsam from various waste scenarios.

*Akanmu & Onyema (2020)* gives valuable open doors to individuals to foster the abilities to take part in the discourse about marine flotsam and jetsam and ways of handling this issue. The Marine Debris Virtual Community Centre (MD-VCC) is made to further develop correspondence and support connected with marine garbage.

*Lyu et al. (2024)* suggest combining numerical models and deep learning to improve monitoring efficiency. A diffusion long short-term memory (LSTM) network is proposed based on the diffusion model's ability to make images and the LSTM network's ability to extract temporal information. The proposed network learns picture evolution from distant sensing data to predict future occurrences. In MODSD dataset experiments, our model beat the convolutional LSTM (ConvLSTM) and generative adversarial network LSTM (GAN-LSTM) time-series image prediction methods. Second, utilise OpenOil to develop a numerical simulation trajectory model. We accommodate for maritime environment differences by calibrating wind drift factors to properly reconstruct oil spill paths. The Sanchi oil leak has a 2500-m error margin. Finally, diffusion LSTM photographs and OpenOil's projected oil spill trajectories were fused to provide short-time interval oil spill scene images, improving monitoring efficiency.

## Sustainable ocean management frameworks

*Han et al. (2020)* present a meta-heuristic whale optimization algorithm (WOA), which assists ships with tracking down a low-energy-utilization and safe course in an enormous scope complex marine climate. A few strategies have been proposed to tackle this issue, however, there are a few weaknesses, for example, no thought of the impact of wind bearing, wind speed and wave. The consequences of our recreation tests show that WOA is more cutthroat than other best-in-class calculations for course arranging.

*Di Luccio et al. (2020)* show that coordinating publicly supported bathymetry information in the work process mathematical model arrangement works on the exactness of the end-product, considering a more itemized spatial circulation example of the ocean ebb and flow driving the Lagrangian tracers.

*Plag, Jones & Garello (2021)* represent the information on the wide open concerning the danger marine flotsam and jetsam to the marine biosphere and humankind stays at low levels. Because of the lack of suitable local area-focused virtual entertainment, it is difficult to work with the framework of an open cross-sectoral local area that may foster greater

collaboration and correspondence among all parties involved in marine rubbish. There is a prototype Marine Debris Virtual Community Centre (MD-VCC) where partners may learn about the numerous sources of marine flotsam and jetsam and the paths waste enters the marine ecosystem.

*Seydi et al. (2021)* grew new oil spill detection (OSD) system in view of a profound learning calculation for optical remove sensing (RS) symbolism. The proposed strategy depended on a multiscale multi-layered leftover part convolutional neural network. The proposed technique explored the profound elements by the two-layered multiscale leftover blocks and, then, at that point, used them at one-layered multiscale remaining blocks. Overall, the suggested technique has an accuracy rate of over 95% and a miss location and deception rate of less than 5%, suggesting its significant potential for OSD. In addition, it was found that the proposed method would be wise to implement compared to other OSD estimates that were examined in this research.

*Xue et al. (2021a)* and *Xue et al. (2021b)* lay out an effective remote ocean flotsam and jetsam identification technique with fast utilizing profound learning strategies. Initial, a genuine remote ocean garbage location dataset (three dimensional dataset) is laid out for additional exploration. Material, fishing net and rope, glass, metal, normal rubbish, elastic, and plastic are all included in the dataset. Another option is the ResNet50-YOLOV3 remote ocean flotsam and jetsam location system. The recognition cycle of distant ocean debris also includes eight high-level identification models. Finally, testing are performed to verify the exhibit of ResNet50-YOLOV3. These trials also show the relevance and feasibility of ResNet50-YOLOV3 in detecting ocean flotsam and jetsam from a distance.

*Ju, Niu & Zhang (2023)* propose a five-parameter polar coordinate dense regression detector (FPDDet) that uses just a centroid, a mapped polar diameter, and two polar angles to solve the overlap and border discontinuity problems with horizontal bounding box (HBB) and oriented bounding box (OBB). We also propose a more suitable dense regression heatmap loss function and a dense regression strategy that dynamically assigns ship target samples in response to ship targets' large-scale variation in SAR images using a covariance-adaptive rotated Gaussian heatmap. A feature enhancement (FE) module that improves target features and reduces background interference will also handle SAR pictures' severe noise pollution. Experimental results show that our FPDDet outperforms the state-of-the-art on the rotating SAR ship detection dataset (RSSDD) and rotated ship detection dataset (RSDD). Following these modifications, mAP is 1.2% and 1.71 percent higher than before.

*Duarte & Azevedo (2023)* use Sentinel-2 data to distinguish plastic litter from driftwood, seaweed, marine snot, sea foam, and pumice. We use manual satellite image interpretation and aggressive gradient boosting learnt on published data. Two Sentinel-2 spectral bands and seven spectral indices are used to teach the technique. We used the project-specific database initially. We realise that ground-truth validation is needed despite the 98% accuracy in categorising possibly dangerous plastic waste. Second, a Wasserstein generative adversarial network generates synthetic data to augment the training dataset. A synthetic-data-only supervised model identified plastic pixels with 83% accuracy. The third is an

ensemble model that assesses classifier prediction uncertainty. We correctly identified 75% of plastic pixels.

*Zhu et al. (2023)* used SVM to model nitrate computation. Orthogonal projection to latent structures support vector machine (OPLS-SVM) model was used to analyse seawater samples from the Western Pacific, Aoshan Bay in Qingdao, China, the South China Sea, and the Yellow Sea. Variable nitrate and turbidity spikes were used. The results showed that the SVM calculation model with the OPLS correction technique improved nitrate measurement precision. An RMSE of 0.22 μmol/L and an R2 of 0.999 were observed in a prediction performance with a temperature range of 5–25 °C and turbidity range of 0-50 NTU. It accurately measures turbidity seawater nitrate concentration, making the presented approach ideal for *in situ* nitrate measurements in challenging seawater conditions.

*Chen et al. (2024)* designed an IoT-based mobile spectral sensing device to collect soil data from several places. A collaborative training model was constructed by merging a fuzzy partial least square (fPLS) model with a large learning network to handle dynamic sensing data. The scalable pseudo input layer with configurable training linking weights generates BLN output feature variables as the number of fuzzy rules is changed. Using a decentralised sensing system and fusion modelling framework, the experiment optimises a model for heavy metals in soil samples detected by portable NIR spectroscopy. The system outperformed the typical PLS model in training and evaluating dynamic sensing data for predictions, according to two components.

*Wang et al. (2024)* present a neurodynamics-driven prediction model for coastal water quality evolution (NDPM-CWQ) after considering tough problems. First, spatiotemporal data is used to train an event-driven deep belief network. Second, we use EDBN model input variable sensitivity analysis to rate triggering factor inversion and traceability and determine spatiotemporal variable effects on water quality. Third, we use Markov chain decision stability and stationary distribution to assess the training EDBN's convergence. Finally, real-world data show that the NDPM-CWQ enhances prediction performance and quantifies triggering component inversion and traceability.

Table 1 show the summary of existing works as below.

## DATASETS

The following is the list of the datasets being considered.

### Coastal pollution dataset

This data represents some of the data gathered in a year of research, on three coastal areas which represent a pollution gradient. The training data consists of only numeric columns. The summary of the columns is shown below:

Table 2 depicts all column of the prescribed dataset. For every feature or predicator, it shows its logical type, storage type, minimum value, mean of these values, maximum value, standard deviation (std), unique values and frequency of mode. The target variable for this research work is being set as the pollution level (three levels, when 0 is clean, 1 is polluted and 2 is the most polluted) and the analysis should divide the data into test and train data

**Table 1  Related works summary.**

| S. No. | Author | Model | Dataset | Performance remarks |
|---|---|---|---|---|
| 1 | _Mercier & Girard-Ardhuin (2005)_ | Support Vector Machines | Spaceborne Synthetic Aperture Radar (SAR) images | Efficient in an operational context |
| 2 | _Li & Wang (2011)_ | Random Forest | Sea Water in Tianjin | 50% Accuracy |
| 3 | _Hong, Fulton & Sattar (2020)_ | Variational Autoencoder | J-EDI (JAMSTEC E-Library of Deep-sea Images) dataset | 0.92 Precision & 0.92 Recall value |
| 4 | _Wu et al. (2020)_ | SVM | DJI (DJI GO 4) dataset | 85.71% Accuracy |
| 5 | _Ronci et al. (2020)_ | CNN | SAR images over Mediterranean Sea | 97.3% Accuracy |
| 6 | _Hipolito et al. (2021)_ | COCO | Data Repository for the University of Minnesota (DRUM) | 98.89% Accuracy |
| 7 | _Kankane & Kang (2021)_ | Mask R-CNN | KaKaXi Camera on beach site both | 87.6% Accuracy |
| 8 | _Rostam et al. (2021)_ | Long Short-term Memory | Tolo harbour dataset | 0.026 MAE |
| 9 | _Kruk, Paturej & Obolewski (2021)_ | Support Vector Machines | Polish Baltic Sea coast | 83.5% Accuracy |
| 10 | _Sasaki et al. (2022)_ | Gaussian Naive Bayes | WolrdView-2 and 3 satellite images | 88% Accuracy |

manually. Figure 1 show the relative feature importance for applying the machine learning model in coastal pollution assessment as below:

## Underwater trash dataset

From the J-EDI marine debris dataset, we obtained this information. That dataset has a wide range of video quality, depth, objects in the scene, and camera settings. They show a wide range of maritime detritus in various levels of decay, opacity, and overgrowth as they were photographed in real-world locations. The water's purity and light's quality might vary greatly from one video to the next. This dataset consists of 5,700 photos which were extracted from the processed movies.

The ultimate objective is to build onboard garbage detecting technologies that are both efficient and accurate. This information is being made public in the hopes that it will help the marine robotics community work toward a solution to the pressing issue of autonomous garbage identification and removal.

## METHODS

There are two major goals of our research: developing a system that can identify Underwater Trash on the coastal and underwater. Research tools were used to gather and evaluate data from diverse sources (_Han & Hong, 2023_; _Ilias et al., 2023_; _Lilhore et al., 2024_). FCN and two-step clustering algorithms are employed as our study framework. Clustering is a highly effective method for detecting similarities across various groups or clusters. According to

**Table 2  Coast pollution dataset.**

| Name | Logical_type | Storage _type | Min | Mean | Max | Std | Unique | Freq of mode |
|---|---|---|---|---|---|---|---|---|
| Month | Numeric, categorical, catlabel, ohe_categorical | int | 1.000 | 6.598 | 18.000 | 4.994 | 7 | 42 |
| Season | Numeric, categorical, catlabel, ohe_categorical | int | 1.000 | 2.260 | 4.000 | 1.023 | 4 | 84 |
| Shore | Numeric, categorical, catlabel, ohe_categorical | int | 1.000 | 2.005 | 3.000 | 0.816 | 3 | 74 |
| Pollution Level | N/A | int | 0.000 | 1.009 | 2.000 | 0.818 | 3 | 74 |
| Sample number | Numeric, categorical, catlabel, ohe_categorical | int | 1.000 | 6.347 | 14.000 | 3.796 | 14 | 21 |
| Organic matter% | Numeric | Real | 0.121 | 0.453 | 1.868 | 0.228 | 202 | 1 |
| Mean Number of Nematode species 1 per gram soil | Numeric | Real | 0.000 | 6.367 | 25.052 | 6.078 | 214 | 3 |
| Mean Number of Turbillaria per gram soil | Numeric | Real | 0.000 | 0.173 | 3.114 | 0.294 | 120 | 93 |
| Mean Number of foraminefera per gram soil | Numeric | Real | 0.000 | 15.071 | 73.600 | 14.411 | 55 | 24 |
| Mean Number of Nematode species 2 per gram soil | Numeric, categorical, catlabel, ohe_categorical | Real | 0.000 | 6.912 | 38.978 | 8.126 | 39 | 35 |
| Water pH | Numeric, categorical, catlabel, ohe_categorical | Real | 7.801 | 8.184 | 8.340 | 0.106 | 20 | 24 |
| Soil pH | Numeric, categorical, catlabel, ohe_categorical | Real | 7.800 | 8.237 | 8.599 | 0.201 | 16 | 28 |
| OC | Numeric, categorical, catlabel, ohe_categorical | Real | 6.450 | 7.996 | 9.300 | 0.898 | 20 | 14 |
| Water Salinity | Numeric, categorical, catlabel, ohe_categorical | Real | 25.400 | 37.355 | 39.600 | 3.146 | 14 | 50 |
| Soil Salinity | Numeric, categorical, catlabel, ohe_categorical | Real | 5.000 | 9.830 | 14.000 | 2.483 | 12 | 34 |
| P | Numeric, categorical, catlabel, ohe_categorical | Real | 21.064 | 28.523 | 35.122 | 3.400 | 22 | 14 |

**Table 2** (*continued*)

| Name | Logical_type | Storage _type | Min | Mean | Max | Std | Unique | Freq of mode |
|---|---|---|---|---|---|---|---|---|
| Total dissolved solids | Numeric, categorical, catlabel, ohe_categorical | Real | 37,800.000 | 56,036.098 | 59,000.000 | 4,622.289 | 17 | 24 |
| PP | Numeric, categorical, catlabel, ohe_categorical | Real | 196.300 | 238.344 | 307.000 | 27.825 | 19 | 14 |
| Conduction | Numeric, categorical, catlabel, ohe_categorical | Real | 39,500.000 | 56,280.822 | 59,000.000 | 4,353.022 | 19 | 21 |
| ORP | Numeric, categorical, catlabel, ohe_categorical | Real | −90.200 | −79.268 | −61.400 | 9.451 | 21 | 14 |
| Specific resistance | Numeric, categorical, catlabel, ohe_categorical | Real | 16.940 | 17.976 | 25.700 | 1.817 | 20 | 14 |
| Temp | Numeric, categorical, catlabel, ohe_categorical | Real | 14.700 | 23.879 | 31.180 | 5.427 | 21 | 14 |
| Conductivity | Numeric, categorical, catlabel, ohe_categorical | Real | 1.840 | 3.020 | 5.235 | 0.942 | 15 | 14 |
| H | Numeric, categorical, catlabel, ohe_categorical | Real | 0.058 | 0.149 | 0.260 | 0.069 | 15 | 14 |
| C-A | Numeric, categorical, catlabel, ohe_categorical | Real | 0.143 | 3.872 | 12.950 | 4.115 | 10 | 14 |
| C-B | Numeric, categorical, catlabel, ohe_categorical | Real | 0.000 | 4.110 | 16.184 | 5.229 | 8 | 42 |
| C-C | Numeric, categorical, catlabel, ohe_categorical | Real | 0.000 | 5.291 | 20.296 | 6.498 | 9 | 28 |

the features of plastic pollution (space, shape, *etc.*), these algorithms examine the hidden information in our research.

## Two-step clustering

To accommodate both categorical and continuous data, the two-step cluster analysis method employs a probability distance metric, predicated on the assumption that cluster model variables are unrelated to one another. In addition, we assume that the distribution of each continuous variable is Gaussian, and that of each categorical variable is multinomial. In-house empirical testing suggests the process holds up well in the face of departures from the independence and distributional assumptions; however, it's important to keep an eye on how well those assumptions are being satisfied (*Memon et al., 2024*; *Radulescu et al., 2024*).

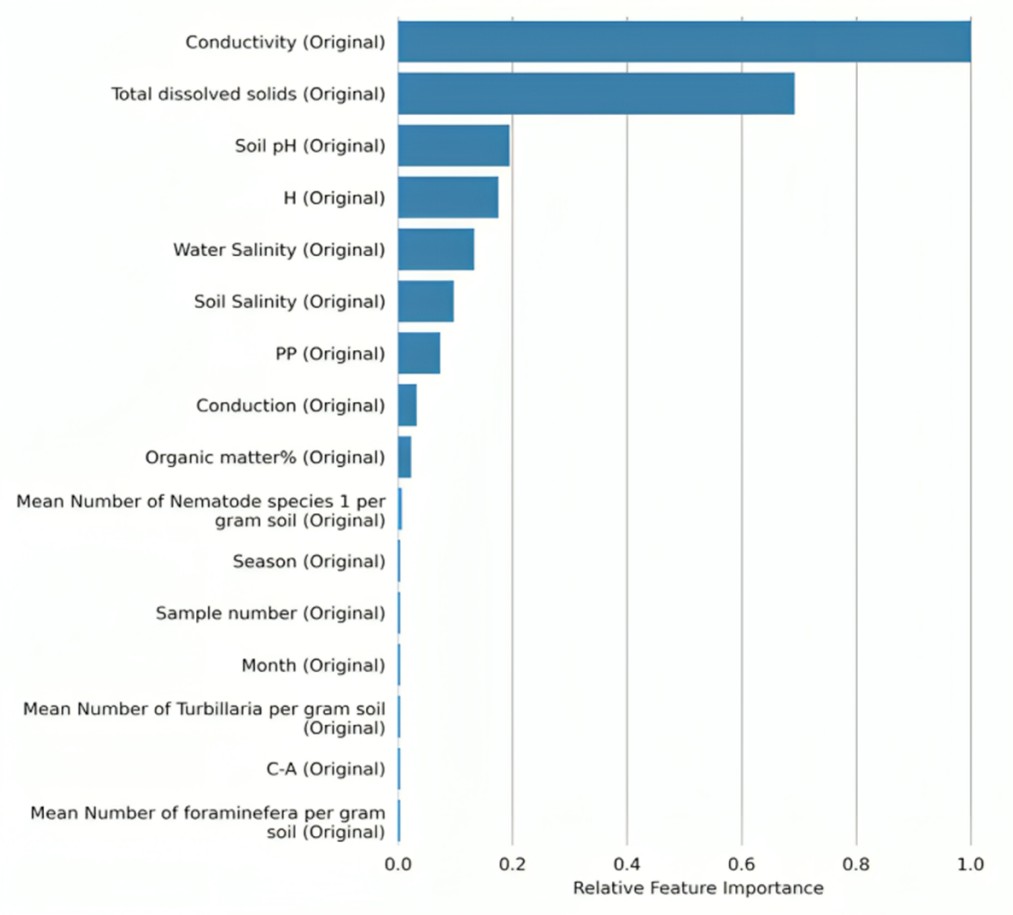

**Figure 1**  Relative feature importance.

The two-step cluster analysis procedure's algorithm may be summed up as follows, covering its two steps:

Step 1.     The first stage of the process involves building a cluster features (CF) tree. The first case is stored in a leaf node at the tree's root that has variable information about that case. Each new example is compared to the preceding cases using the distance measure to determine whether it should be added to an existing node or should form a new node. An aggregate of case-specific variables is stored in a node that contains many cases. Because of this, the CF tree may be seen as a condensed version of the entire data set.

Step 2.     Following this, an agglomerative clustering technique is used to classify the CF tree's leaf nodes. It is possible to get various answers by employing agglomerative clustering. Each of these cluster solutions is evaluated using either Schwarz's Bayesian Criterion (BIC) or the Akaike Information Criterion (AIC) to identify the ''optimal'' number of clusters.

In the second stage, TwoStep can employ hierarchical clustering to analyze the data and identify the best possible number of clusters to use. The results of a hierarchical clustering process may be seen immediately, with a series of divisions such as 1, 2, 3, …, clusters. On the other hand, the sequence could only be generated by running the k-means algorithm an arbitrary number of times (one for each cluster) (*Schubel, 1994*; *Silhadi, Refes & Mazouzi, 2020*; *Thiel et al., 2018*).

Finding plastic trash in the water can be difficult, but a two-step clustering strategy can help. The usefulness of two-stage clustering in this scenario is discussed below.

### *Data collection and preprocessing*
In order to get the dataset ready for clustering, preprocessing is performed. Images may need to be resized or normalized, noise or unwanted parts removed, and useful attributes extracted before they can be put to use.

### *Extracting features*
Features representative of plastic trash should be collected from the photos in the first step of the two-stage clustering method. Color histograms, texture descriptors like Gabor filters or local binary patterns, shape descriptors like contours or geometric features, or any combination thereof, are all examples of such features.

The traits selected are those most able to differentiate marine trash made of plastic from other things in the water. Choose these attributes to accurately capture the appearance of plastic trash in the provided images.

## Initial clustering
Clustering algorithms like k-means and hierarchical clustering are used to produce an initial grouping on the dataset in the first part of the two-step clustering process. The goal of this process is to use the collected characteristics to create first clusters within the dataset.

When sorting plastic trash for the first time, like items are gathered together. This process aids in locating groupings of pictures that may include plastic trash.

### *Clarification and categorization*
The second phase, following the first clustering, is to refine the clusters and classify the pictures more precisely. Density-based clustering and model-based clustering are two common alternatives used for this stage.

Images are properly categorized into groups by the enhanced clustering algorithm, which takes into account criteria like proximity, density, and similarity metrics. This aids in the identification of plastic trash amongst other ocean debris.

### *Analyses and verifications*
Finally, the effectiveness of the two-stage clustering method is assessed in terms of finding plastic trash. Precision, recall, F1-score, and clustering validity indices are some examples of quantitative measurements that may be used for this purpose. Clusters of plastic trash can also be evaluated for quality and accuracy through visual inspection and human confirmation (*Vikas & Dwarakish, 2015*).

The method employs a two-step clustering process, which allows it to accurately recognize and categorize plastic trash based on visual similarities. This aids in the detection and categorization of ocean plastic debris, which is crucial for further research and mitigation efforts. Two-step clustering can be an effective approach for detecting plastic garbage in the ocean environment. By employing two-step clustering, the approach can effectively identify and group similar instances of plastic garbage based on their visual features. This helps in detecting and categorizing plastic waste in the ocean environment, providing valuable insights for further analysis and mitigation efforts. It's important to note that the success of this approach relies on the quality and representativeness of the dataset, as well as the selection and extraction of relevant features. Continuous improvement and refinement of the clustering algorithm and feature selection process can enhance the accuracy and reliability of plastic garbage detection in the ocean environment. It's worth noting that the success of FCN-based plastic garbage detection depends on factors such as the size and quality of the training dataset, the choice of FCN architecture, and the effectiveness of data augmentation techniques. Continuous refinement and improvement of the FCN model can enhance its performance in accurately identifying and delineating plastic garbage in the ocean environment.

By leveraging the capabilities of FCN, this approach enables pixel-level segmentation of plastic garbage, providing valuable insights for monitoring and addressing plastic pollution in oceans. The effectiveness of this method is dependent on the dataset's quality and representativeness, as well as the characteristics chosen and extracted. The effectiveness and dependability of plastic rubbish identification in the water may be improved by enhancing the clustering algorithm and the feature selection process on an ongoing basis.

## YOLO model

A real-time object identification technique known as YOLO stands for "You Only Look Once". All of YOLO's layers are convolutional, creating a neural network (FCN). Using skip connections and upsampling layers, there are 75 convolutionary levels in this algorithm. A convolution layer with a stride of 2 is utilized to down sample the feature maps without the usage of pooling. It learns from entire photos and improves detection performance by doing so directly. In comparison to other object-detection algorithms, this model offers a few advantages.

- YOLO is very quick.
- YOLO takes the whole image while training and test time so it completely encodes related data about classes and their appearance.
- YOLO learns generalized representations of objects and thus outperforms other detection methods.

The object detection includes seeing certain things in digital photos or movies. Humans, automobiles, chairs, stones, structures, and even animals have all been spotted. One effective deep learning architecture that may be used to find trash in the ocean is the FCN.

YOLO treats real-time object detection as a regression issue. Unlike Faster R-CNN, YOLO proposes and classifies regions in one network pass. YOLO's first equation grids

input picture S * S. Each cell predicts B bounding boxes, confidence scores, and C item class probabilities. The biggest benefit is speed: Designed for real-time applications, YOLO processes photos in one forward pass. Grid division may be too coarse to capture minor item details, making it hard to handle.

$$\text{Predictions per grid cell} = B*5+C. \tag{1}$$

$B$ is bounding boxes, 5 is $x, y, w, h$ coordinates and confidence score, and C is classes. This approach lets YOLO forecast numerous items per grid cell but suffers with overlapping objects. With four parameters—box center coordinates $(x, y)$, width $w$, and height $h$—YOLO predicts bounding boxes. Normalized by image size:

$$x = \sigma\left(t_x\right) + c_x \tag{2}$$

$$y = \sigma\left(t_y\right) + c_x \tag{3}$$

$$w = p_w e^{t_w} \tag{4}$$

$$h = p_h e^{t_h} \tag{5}$$

where $t_x$, $t_y$, $t_w$, $t_h$, are the network's raw predictions, $c_x$, $c_y$ are the coordinates of the op-left corner of the grid cell, and $p_w$, $p_h$ are predefined anchor box dimensions. These equations ensure that the predicted bounding boxes are within a reasonable range relative to the anchors.

YOLO employs anchor boxes to predict bounding boxes relative to predefined shapes, which helps the model generalize across various object scales and aspect ratios. A significant part of YOLO's object detection mechanism involves computing the Intersection over Union (IoU) between predicted and ground truth boxes to determine the quality of predictions:

$$\text{IoU} = \frac{\text{Area of Overlap}}{\text{Area of Union}}. \tag{6}$$

Area of Overlap/Area of Union (IoU) compares the expected and actual bounding boxes. IoU boosts alignment, while YOLO increases it during localization training. The key benefit of IoU is measuring localization quality well. IoU optimization may not help categorization, especially when object sizes fluctuate substantially between pictures.

A potential strength of YOLO is its image digestion speed. Grid-based object localization and direct bounding box regression simplify real-time predictions. The IoU metric and anchor boxes make the model adaptable to different item sizes and shapes, boosting localization accuracy. The single-stage YOLO pipeline localizes and classifies objects for efficiency. However, this strategy has limitations. YOLO anticipates several objects per grid

cell, therefore it may overlook minor things in busy surroundings. The grid may also limit the model's ability to recognize objects across cells, decreasing large object performance. YOLO's grid-based prediction algorithm is rapid but may underestimate small or dense objects in grid cells. The architecture implies each grid cell can recognize one object, which can limit small object scenarios. Anchor boxes help YOLO, but high item density persists. Localization, confidence, and classification losses in YOLO's loss function can also generate an imbalance, especially when object size varies greatly across the dataset. Smaller items may struggle because larger ones dominate localization loss. Compared to Faster R-CNN, YOLO is efficient but not precise enough to distinguish small and overlapping objects.

### Collecting and formatting data sets

Gathering photos of the ocean's surroundings is the first step in training an FCN to recognize plastic trash. Images like this should show examples of plastic trash in addition to other parts of the ocean ecosystem. It is crucial that the dataset contains enough samples of plastic rubbish of varying shapes and sizes, and that it covers a wide variety of maritime areas. Images of plastic trash should have their respective areas or pixels appropriately identified in the dataset. The FCN is trained using these annotations as the gold standard.

### Design and instruction

Since the FCN architecture was developed for picture segmentation, it may be used to spot trash floating in the ocean. The FCN is made up of an encoder and a decoder; the former takes an input picture and extracts high-level features, while the latter creates a segmentation map at the pixel level. Supervised learning methods are frequently used to train the FCN. The network is trained using the labeled dataset, with the parameters optimized such that it can successfully segment and recognize patches of plastic waste within the photos. This is achieved by reducing the amount of variation between the FCN's projected segmentation maps and the actual annotations. To improve the FCN's capacity to generalize, data augmentation techniques can be used during training to increase the variety of the training samples.

### Detection and inference

Once the FCN has been trained, it may be used to identify pieces of plastic debris hidden inside photographs of the ocean's surface. The FCN uses forward propagation to create a segmentation map at the pixel level from an input picture. Each pixel is classified as either plastic waste or other things based on the label or probability assigned to it by the segmentation map. To improve the precision of the plastic rubbish zones, the segmentation map might be post-processed using thresholding or morphological treatments.

### Analyses and verifications

Metrics such as Intersection over Union (IoU), accuracy, recall, and F1-score may be used to assess the FCN's effectiveness in the detection of plastic trash. Predicted segmentation maps are compared to the annotated ground truth in order to determine how closely they match. The quality of the plastic rubbish detection zones may also be evaluated by visual examination and professional evaluation. It is important to remember that the performance

of FCN-based plastic waste identification is influenced by a number of variables, including the amount and quality of the training dataset, the selected FCN architecture, and the efficiency of data augmentation methods. The FCN model can do a better job of precisely detecting and delimiting plastic rubbish in the ocean if it is continuously refined and improved.

This method, which makes use of FCN's capabilities, allows for pixel-level segmentation of plastic rubbish, yielding useful information for tracking and preventing marine debris.

## PROPOSED METHODOLOGY

Two fundamental concerns are being addressed by this phenomenon:

- What is the item? This question asks the users to name the thing the users see in a particular picture.
- Where did the users find it? With this question, they are trying to pin down exactly where in the image the object is located.

The working of YOLO is explained with the help of flowchart in Fig. 2. The pseudocode for YOLO is given underneath.

**YOLO Algorithm**

Step 1: Choose the box with the maximum objective function.

Step 2: Next, compare the overlap (Intersection Over Union, IOU) of the chosen box with other boxes. IOU is calculated as:

$$\text{Intersection of union} = \frac{\text{Area of overlap}}{\text{Area of union}} \qquad (7)$$

Step 3: Eliminate the bounding boxes with overlap > 50%.

Step 4: Shift to the next highest objectiveness score.

Step 5: Lastly, repeat steps 2–4.

Detection methods include R-CNN, Retina-Net, and Single-Shot MultiBox Detector. Object detection (SSD). A single algorithm run is unable to find things, even though these techniques have overcome the limitations of limited data and model-based object recognition. The YOLO algorithm has become popular because it outperforms the other object detection methods.

### Tuning the learning rate hyper-parameter of YOLO algorithm with Bayesian optimization

However, depending on the optimizer's tuning choices, an improvement over YOLO models is more noticeable. Image of input size, number of epochs, batch size, learning rate, momentum, and activation function are all factors that may be tweaked to enhance YOLO's performance. To further cut down on training time and boost model performance, tweaking hyper-parameters like learning rate and momentum inside the training algorithm is highly recommended. Poor hyper-parameter tuning would lead to under- or over-fitting of the model. Increase regularization, quicken the rate of training, and introduce instability.

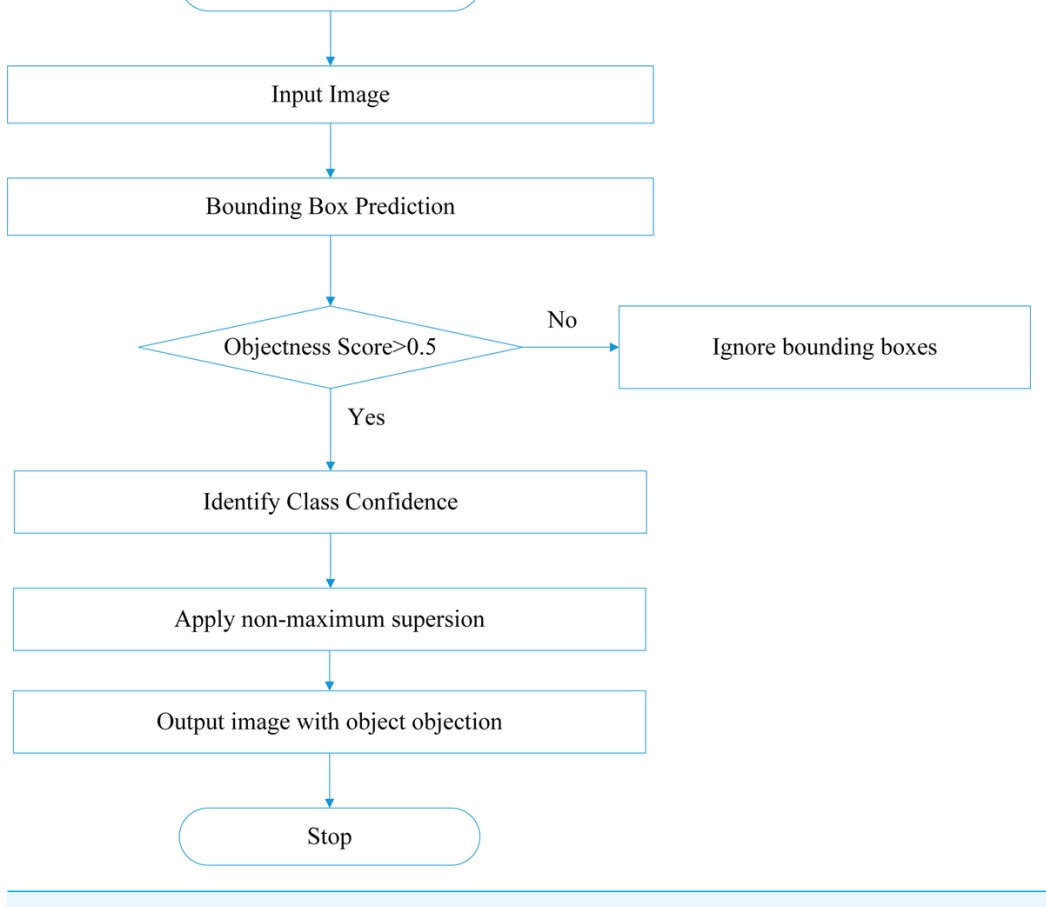

**Figure 2  Flowchart of YOLO model.**

Consequently, the purpose of this research was to fine-tune the optimizer, learning rate, and momentum of the YOLO model so that it could be used effectively for detection in the deep sea. YOLOv5 was chosen for the model optimization in this investigation for a number of reasons.

- The state-of-the-art target detection technique using two content security policy (CSP) structures (CSP1 X and CSP2 X) to extract generic characteristics specifically in underwater picture.
- This model is capable of dynamically adjusting the network's depth and breadth in response to the volume of data being processed (self-adaptation to small underwater objects).
- This model can provide the best possible training results and detection accuracy, as demonstrated in this research.

In contrast to random and grid search, Bayesian techniques remember previous evaluations and utilize them to build a probabilistic model that maps hyperparameters to the likelihood of a score on the objective function.

As $p(y|x)$, this model is a "surrogate" for the objective function in the relevant literature. Choosing the hyperparameters that perform best on the surrogate function allows Bayesian approaches to determine the next set of hyperparameters to assess on the real objective function, which is significantly more difficult to optimize than the surrogate function. What this means, in words:

- Construct a probabilistic stand-in for the goal function.
- Determine the optimal values for the surrogate's hyperparameters.
- Use these settings as hyperparameters for the actual goal function.
- Revise the proxy to account for the latest findings.
- Continue iterating through steps 2–4 until the time limit or maximum iterations are reached.

These methods implement the Bayesian tenet of "less incorrect with more data" by iteratively refining the surrogate probability model in response to each new assessment of the objective function.

Bayesian optimization (BO) and transfer learning (TL) for hyperparameter change in YOLO models improve object detection accuracy and efficiency. The fast and accurate deep learning system YOLO is popular for real-time object detection. Optimizing performance requires fine-tuning its hyperparameters, which greatly impact detection accuracy and processing efficiency. BO optimizes complex functions with few evaluations, ideal for this. TL uses pre-trained models to adapt YOLO to a similar task, speeding training and improving performance with less labeled data. BO uses a Gaussian probabilistic model to approximate the goal function that links hyperparameter settings to model performance. By exploring intriguing search space regions, BO intelligently selects hyperparameters, unlike brute-force grid search or random search. BO optimises YOLO hyperparameters including learning rate, batch size, and momentum while minimizing model assessment computing cost per iteration. This method works for YOLO models, which have many hyperparameters and require a lot of computing power to train.

TL allows YOLO models use knowledge from a COCO or ImageNet-trained network. By fine-tuning a pre-trained YOLO model on a smaller dataset, TL reduces initial training time and resources. TL provides a stable baseline model with optimal weights, reducing hyperparameter tweaking evaluations. TL and BO interact to make training more efficient by fine-tuning only a few hyperparameters to increase target dataset performance. Staged YOLO hyperparameter optimization combines BO and TL. Select and transfer a pre-trained YOLO model to the new task using TL. Bayesian Optimization adjusts transferred model hyperparameters for the target domain. Because the pre-trained model has a strong starting point and BO can quickly determine the optimal hyperparameter settings, this dual method speeds convergence. TL reduces the model's reliance on new data, while BO maximizes generalization with carefully adjusted hyperparameters to prevent overfitting, a common deep learning problem with smaller datasets. Hyperparameter adjustment by BO for TL reduces training runs. BO optimises important parameters for better results with fewer assessments. YOLO, a computationally expensive model, benefits from this because each training iteration requires time and resources. TL initializes the model well, so BO can

adjust learning rate and regularization parameters for the new dataset. Combining them optimizes models efficiently and robustly.

YOLO hyperparameter adjustment with BO and TL improves object detection. BO intelligently navigates the hyperparameter space to find the best settings, whereas TL uses a pre-trained model to speed up training and improve accuracy. This hybrid method works effectively in domains with few training resources or fine-tuning datasets. Merging BO and TL in YOLO models achieves state-of-the-art object detection in real-world applications in a realistic, efficient manner.

```
# Step 1: Import necessary libraries
import numpy as np
import bayes_opt # Bayesian Optimization library
import torch
import torchvision.transforms as transforms
from bayes_opt import BayesianOptimization
from yolo_model import YOLO # Assume a YOLO model class is defined elsewhere
# Step 2: Define the data loading and preprocessing steps for marine pollution dataset
def load_marine_pollution_data():
# Load the dataset (train, validation, and test splits)
train_data = # Load training data
val_data = # Load validation data
test_data = # Load test data
# Define any necessary transformations for images
transform = transforms.Compose([
transforms.Resize((416, 416)), # Resize to YOLO input size
transforms.ToTensor(),
# Add more transformations if necessary
])
# Apply transformations to the dataset
train_data = transform(train_data)
val_data = transform(val_data)
test_data = transform(test_data)
return train_data, val_data, test_data
# Step 3: Define the function to train and evaluate the YOLO model
def train_yolo_model(train_data, val_data, hyperparams):
# Unpack the hyperparameters to be optimized (learning rate, batch size, etc.)
lr = hyperparams['learning_rate']
batch_size = int(hyperparams['batch_size'])
momentum = hyperparams['momentum']
# Initialize YOLO model with Transfer Learning (load pre-trained weights)
model = YOLO(pretrained =True)
# Modify last layers for the new marine pollution task
model.modify_output_layers(num_classes =2) # Example: 2 classes (pollution vs no
pollution)
```

```
# Define optimizer (e.g., SGD or Adam)
optimizer = torch.optim.SGD(model.parameters(), lr =lr, momentum =momentum)
# Define loss function (e.g., Binary Cross Entropy or Focal Loss)
criterion = torch.nn.BCELoss()
# Training loop
for epoch in range(num_epochs):
model.train()
for batch in train_data:
images, labels = batch
optimizer.zero_grad()
outputs = model(images)
loss = criterion(outputs, labels)
loss.backward()
optimizer.step()
# Evaluate the model on the validation set
val_loss, val_accuracy = evaluate_model(model, val_data)
# Return negative validation loss (BO tries to maximize, so minimize loss by negating it)
return -val_loss
# Step 4: Define the evaluation function
def evaluate_model(model, val_data):
model.eval() # Set model to evaluation mode
val_loss = 0
correct = 0
total = 0
with torch.no_grad():
for batch in val_data:
images, labels = batch
outputs = model(images)
val_loss + = criterion(outputs, labels).item()
predicted = torch.argmax(outputs, dim =1)
total + = labels.size(0)
correct + = (predicted = = labels).sumitem()
val_loss / = len(val_data)
val_accuracy = correct/total
return val_loss, val_accuracy
# Step 5: Define the objective function for Bayesian Optimization
def bo_objective(learning_rate, batch_size, momentum):
# Define the hyperparameter dictionary to pass into the training function
hyperparams = {
'learning_rate': learning_rate,
'batch_size': batch_size,
'momentum': momentum
```

```
}
# Load the data
train_data, val_data, test_data = load_marine_pollution_data()
# Train the model and return the evaluation metric (negative validation loss)
val_loss = train_yolo_model(train_data, val_data, hyperparams)
return val_loss
# Step 6: Initialize the Bayesian Optimizer with the hyperparameter search space
bo = BayesianOptimization(
f =bo_objective,
pbounds ={
'learning_rate': (1e−5, 1e−2), # Search space for learning rate
'batch_size': (8, 32), # Search space for batch size
'momentum': (0.5, 0.99) # Search space for momentum
},
random_state =42,
verbose =2 )
# Step 7: Run Bayesian Optimization to find the best hyperparameters
bo.maximize(
init_points =10, # Number of random initial points
n_iter =25 # Number of optimization iterations )
# Step 8: Retrieve the best hyperparameters found by BO
best_hyperparams = bo.max['params']
# Step 9: Train the YOLO model using the best hyperparameters on the full training
data
train_data, val_data, test_data = load_marine_pollution_data()
final_model = train_yolo_model(train_data, val_data, best_hyperparams)
# Step 10: Evaluate the final model on the test set
test_loss, test_accuracy = evaluate_model(final_model, test_data)
# Step 11: Output the final results
print(f"Test Loss: {test_loss}")
print(f"Test Accuracy: {test_accuracy}")
```

BO and TL for YOLO hyperparameter tuning offer a novel technique to maximize model performance while reducing computational costs. Grid and random search are computationally expensive and inefficient for complex models like YOLO with enormous hyperparameter space. However, BO is more efficient since it models the objective function as a probabilistic process (typically a Gaussian process) and consciously selects the next hyperparameters to assess using an acquisition function. This reduces model evaluations to find appropriate hyperparameters for computationally demanding models like YOLO. TL reduces training time and data for new tasks by using pre-trained YOLO models on massive datasets like COCO or ImageNet. It employs low-level features learned by YOLO on previous jobs, which are often applicable across datasets. Pre-trained YOLO layers capture broad object detection qualities, but BO-guided hyperparameter tuning

personalized them to the target dataset. Hyperparameters are carefully regulated to retain generalizable knowledge and optimize task-specific performance.

TL directed BO's inclusion in YOLO's hyperparameter fine-tuning, a big advance. To balance speed, accuracy, and generalization, manually tuning YOLO's hyperparameters—learning rate, batch size, and momentum—takes effort and expertise. BO automatically balances exploration (trying new hyperparameter values) with exploitation (perfecting existing good setups). This simplifies and data-efficiently performs well in medical imaging and autonomous driving, where datasets are tiny or domains are specific. Applying YOLO to real-world situations involves model generalization to new data. BO and TL reduce overfitting by optimizing YOLO model hyperparameters. BO's probabilistic approach, which creates a posterior distribution over the objective function, increases generalization by considering hyperparameter uncertainty and performance. Random search ignores uncertainty and may choose poor hyperparameters for complex models. This unique method is generally applicable. YOLO models fine-tuned with BO and TL can be employed in industrial defect identification, wildlife monitoring, and traffic analysis where labeled data is scarce and computational resources are limited. BO optimizes hyperparameters with fewer iterations, decreasing processing overhead, while TL lets the YOLO model swiftly adapt to the new task. This enables YOLO implementation in resource-constrained or specialty domains efficient and scalable, aiding computer vision and deep learning. Finally, BO and TL improve model interpretability and prediction confidence in YOLO hyperparameter tuning. BO calculates uncertainty for each hyperparameter configuration, showing the model's confidence in the parameters. Safety-critical applications like medical diagnosis and driverless vehicles require understanding forecast uncertainty. BO's systematic nature makes tuning replicable and justifiable using probabilistic reasoning, making the YOLO model more precise, clear, and dependable.

## RESULTS

**Experiment 1** Coastal pollution assessment through two-step clustering

The pre-clustering, parsing of typical data kinds, and clustering steps are all performed using the two-step clustering method. It is chosen whether or not to begin a new cluster during pre-clustering after each set of data has been analysed and evaluated. Table 3 highlights the model specifications of two-steps clustering as given below:

This choice is made based on how far apart the data are. The Euclidean distance and the log-probability distance are two distance metrics. Feature importance for this algorithm is being displayed in Fig. 3.

It is common for data that cannot be clustered to be assessed during the data analysis step. The data is segregated as external if the inclusion cannot be achieved despite all attempts to incorporate it. Table 4 highlighted the model quality in the help of Goodness and Importance factor.

In the cluster stage, a tree structure is created. All data starts to be distributed from root to leaves. Table 5 shows the Cluster-1 Profilein term of Within-Cluster Feature Importance.

Each piece of information is connected to a nearby branch. When the ideal number of groups has been obtained, the cluster is linked to another cluster in another branch that

**Table 3** Model specifications.

| | |
|---|---|
| Number of Regular Clusters | 2 |
| Number of Outlier Clusters | 0 |
| Continuous Inputs | Month |
| | Season |
| | Shore |
| | Pollution Level |
| | Organic matter% |
| | OC |
| | Water Salinity |
| | P |
| | Total dissolved solids |
| | PP |
| | Conduction |
| | ORP |
| | Specific resistance |

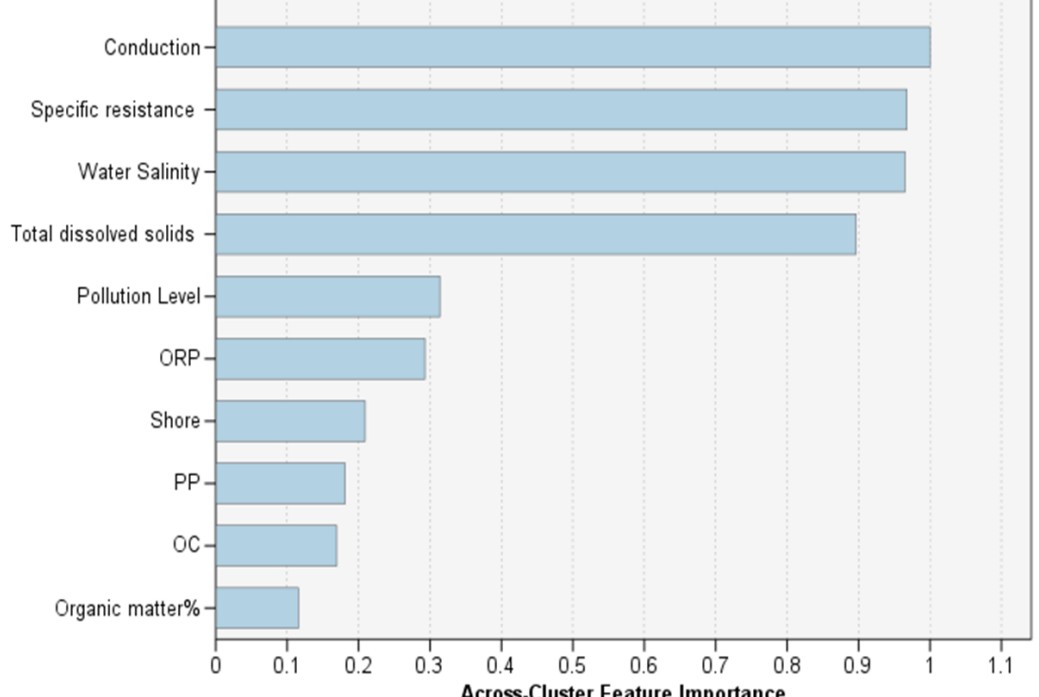

The top 10 inputs are shown.

**Figure 3** Across-cluster feature importance.

meets the distance criterion's requirements. Table 6 show the Cluster-2 Profile in term of Within-Cluster Feature Importance.

To repeatedly decide the appropriate number of clusters, BIC (Schwarz's Bayesian Information Criterion) or AIC (Akaike's Information Criterion) procedures are used.

**Table 4  Model quality.**

| Cluster | Number of records | Goodness | Importance |
|---|---|---|---|
| Cluster-1 | 186 | 0.34 | 1.00 |
| Cluster-2 | 33 | 0.32 | 1.00 |

**Table 5  Cluster-1 profile.**

| Input | Center | Within-cluster feature importance[a] |
|---|---|---|
| Conduction | 0.37 | 1.00 |
| Specific resistance | −0.37 | 0.97 |
| Water Salinity | 0.38 | 0.96 |
| Total dissolved solids | 0.38 | 0.90 |
| Pollution Level | −0.21 | 0.31 |
| ORP | −0.28 | 0.29 |
| Shore | 0.00 | 0.21 |
| PP | 0.19 | 0.18 |
| OC | 0.18 | 0.17 |
| Organic matter% | 0.11 | 0.12 |
| P | 0.05 | 0.11 |
| Month | 0.07 | 0.07 |
| Season | −0.08 | 0.01 |

**Table 6  Cluster-2 profile.**

| Input | Center | Within-cluster feature importance[a] |
|---|---|---|
| Conduction | −2.09 | 1.00 |
| Specific resistance | 1.92 | 0.97 |
| Water Salinity | −2.11 | 0.96 |
| Total dissolved solids | −1.98 | 0.90 |
| Pollution Level | 1.21 | 0.31 |
| ORP | 1.60 | 0.29 |
| Shore | −0.01 | 0.21 |
| PP | −0.98 | 0.18 |
| OC | −0.95 | 0.17 |
| Organic matter% | −0.68 | 0.12 |
| P | −0.27 | 0.11 |
| Month | −0.40 | 0.07 |
| Season | 0.46 | 0.01 |

**Notes.**
Cluster centers show modes for categorical inputs, and means for continuous inputs.
[a] This is the importance of an input to a particular cluster.

**Experiment 2** Underwater Trash Detection with YOLO algorithm

Using test-time augmentations (TTA), we may increase the accuracy of our predictions even further: each picture is enhanced (horizontal flip and 3 different resolutions) and the final prediction is an ensemble of these augmentations.

YOLO is a sophisticated CNN for real-time object identification called a convolution neural network (CNN). YOLO's popularity is due to its ability to operate in real-time and its high level of accuracy.

The following parameters help in achieving better understanding and analysis of model and its performance.

a. Accuracy is given by the formula:

$$Accuracy = \frac{TP + TN}{TP + TN + FP + FP} = \frac{Correct\ predictions}{Total\,predictions}. \tag{8}$$

b. Precision is calculated as:

$$Precision = \frac{TP}{TP + FP} = \frac{Predictions\ actually\ positive}{Total\ predicted\ positive}. \tag{9}$$

c. Recall (TPR, Sensitivity)is calculated as:

$$Recall = \frac{TP}{TP + FN} = \frac{Predictions\ actually\ positive}{Total\ actual\ positive\ and\ actual\ negative}. \tag{10}$$

d. AP (Average precision) is a popular metric in measuring the accuracy of object detectors. It is measured as finding the area under the precision–recall curve above.

It is calculated using the formula:

$$AP = \int_0^1 p(r)\,dr. \tag{11}$$

The following Fig. 4 shows the feature extraction.

In order to fine-tune the model, we may unfreeze it completely and retrain it on our data with an extremely low learning rate. By progressively modifying the pre-trained features to the fresh data, this may be able to produce considerable gains. The following Fig. 5 shows the results of training.

The hyperparameters-configurations file may be used to change the learning rate parameter. Hyperparameters specified in the built-in 'hyp.finetune.yaml' file will be used for the instructional example because they have a considerably lower learning rate than the default. The stored weights from the previous step will be used to re-initialize the weights.

As a convenience for this lesson, we'll utilise the YOLOv5s6 model, which has a relatively modest number of parameters. In this section, we'll discuss some of the most often utilised training methods, as well as a few others. Here, in this section, we'll discuss some of the most often utilised training methods in Fig. 6.

The backbone layers of a model function as a feature extractor, whereas the head layers are responsible for calculating the output predictions. We'll utilise the same backbone as the pre-trained COCO model and simply train the model's head in order to compensate

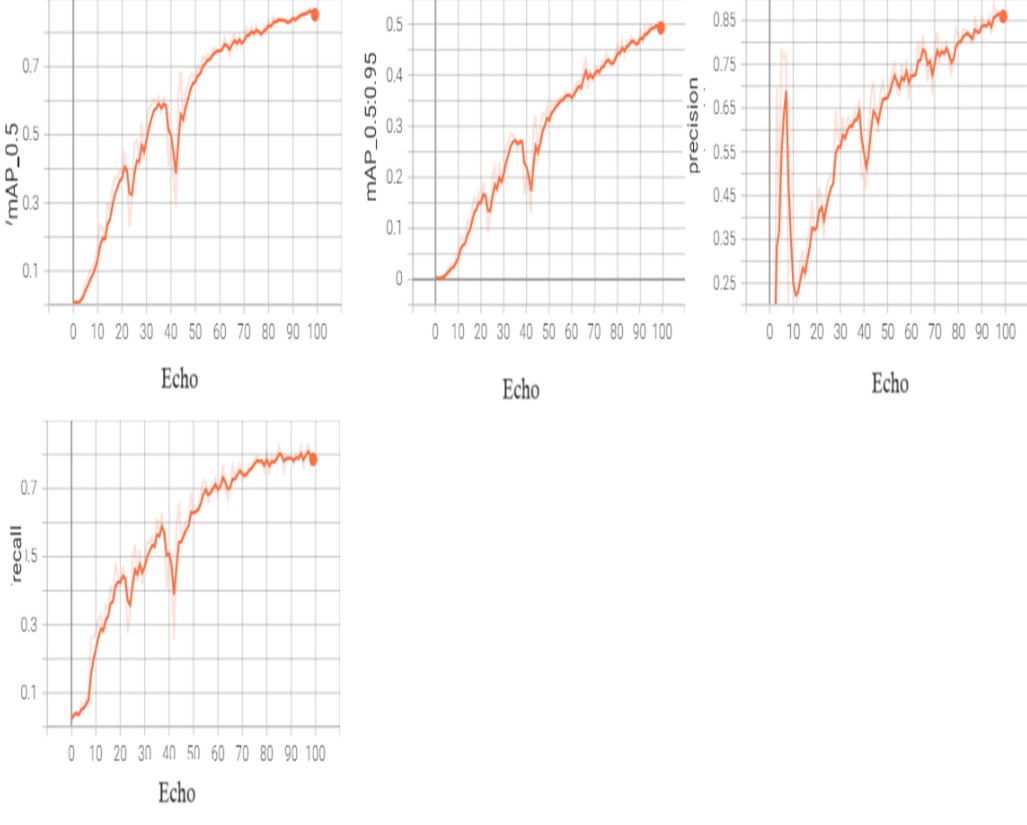

**Figure 4** Inference results.

for a short dataset size. Backbone YOLOv5s6 has 12 layers, and the 'freeze' parameter fixes them all. The confusion matrix created is shown in Fig. 7.

The validation script will be used to test our model. The 'task' option can be used to adjust the divides between the training, validation, and test datasets for evaluating performance. The following Fig. 8 is an evaluation of the test dataset split. The following is an evaluation of the test dataset split:

Multi-class probabilities and bounding boxes may be predicted concurrently using YOLO Precision-confidence curve is shown in Fig. 9.

Recall measures how much of the true bbox was successfully predicted (Real positives/(True positives + True Negatives)). Precision measures how much of the true bbox was accurately predicted. MAP 0.5 is the average precision (mAP) for an intersection over union threshold of 0.5-0.6-0.5. An IoU threshold range of 0.5 to 0.95 is used to calculate the average mAP. Figure 10 shows confidence curve.

It was trained on synthetic data to examine the performance of object detectors and estimate the need for more real data for two-step clustering and FCN, which are the methods discussed in this work. Aside from that, the suggested sensor system's computing needs were studied using FCN. Figure 11 demonstrates the relationship between recall and confidence gained by proposed model.

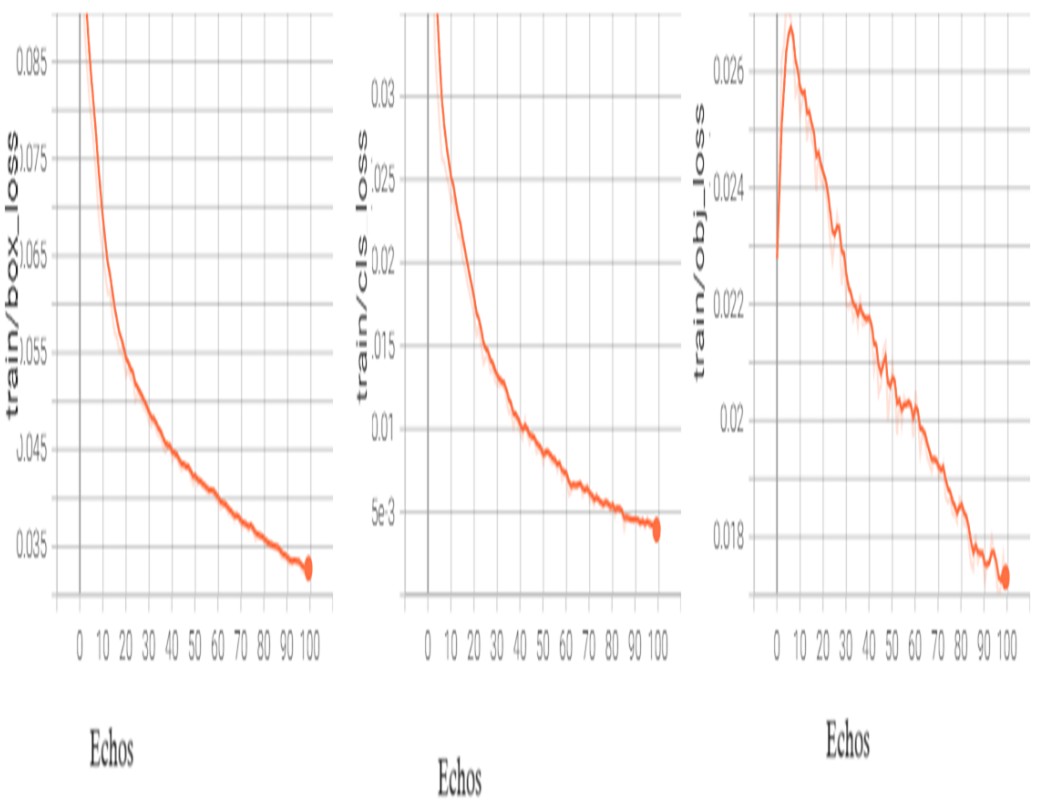

**Figure 5** Training box_loss.

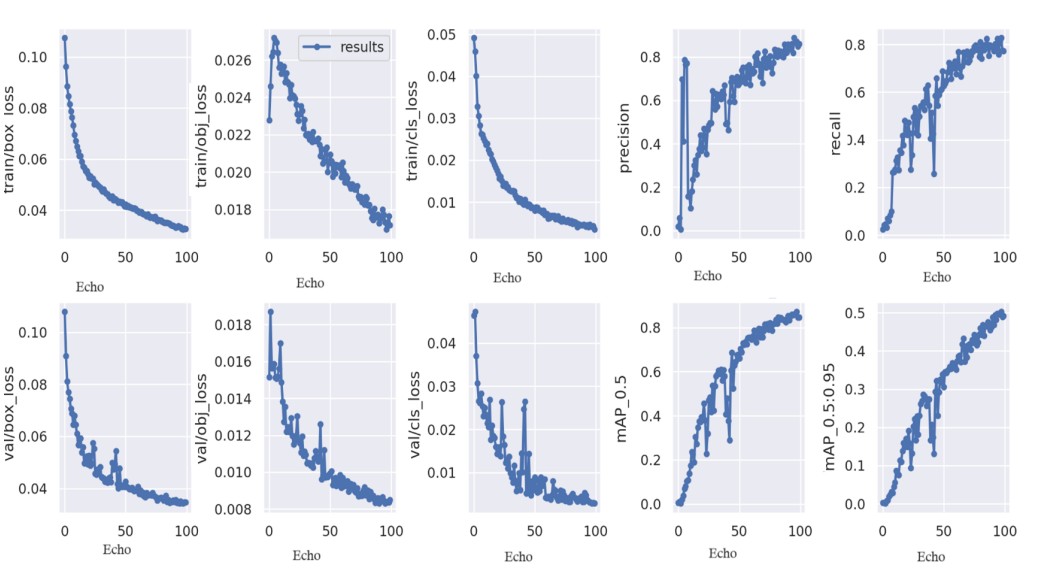

**Figure 6** Result achieved by proposed model.

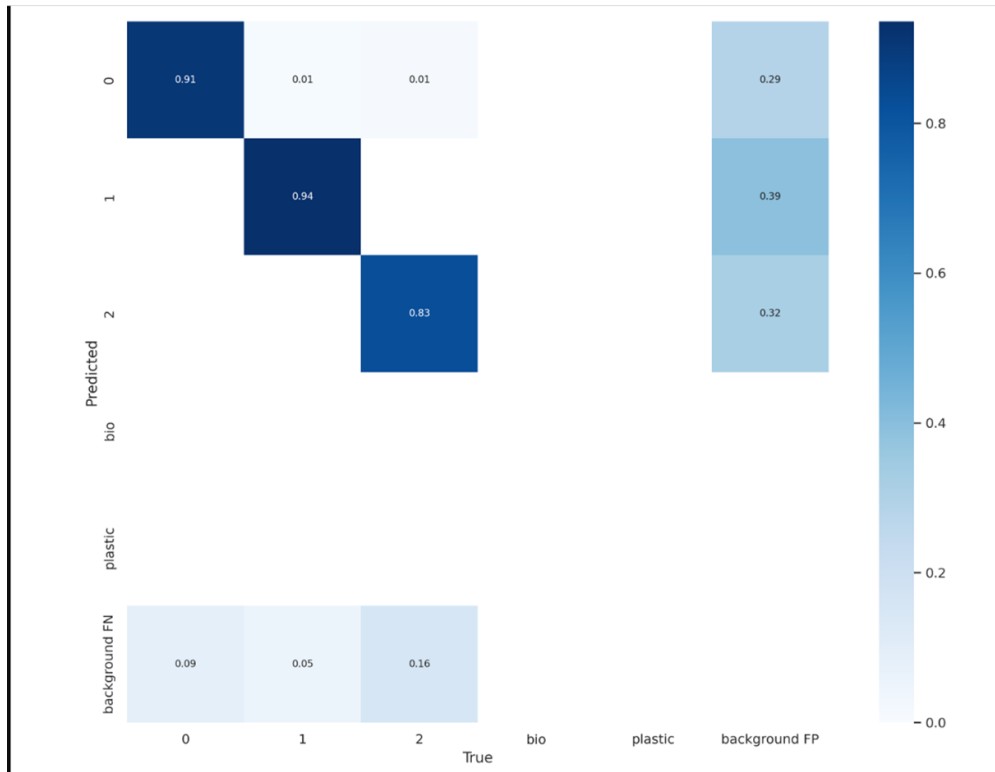

**Figure 7 Confusion matrix.**

## Ablation studies

To further substantiate the contributions of our model, we have conducted comprehensive ablation studies in Table 7. These studies aim to assess the impact of various components and design choices on the overall performance of our model. The ablation studies include:

For the purpose of forecasting marine pollution for long-term ocean health, there are a number of criteria that may be used to evaluate the CNN *vs* the modified YOLO model. Key considerations are discussed below.

## Precision in object detection

Object detection tasks, such as the detection and localisation of marine pollution in photos, are amenable to both the modified YOLO model and CNN. For real-time object identification, the YOLO model has been tweaked into YOLOv3 or YOLOv4, both of which have shown to be highly effective. However, convolutional neural networks (CNNs) are a type of deep learning architecture that may be utilized for both image categorization and object recognition.

The modified YOLO model is well-known for its rapid picture processing and precise predictions in the field of object recognition. It can identify marine pollution of varying sizes and shapes because of its multi-scale and multi-granular detection capabilities. Similarly to the modified YOLO model, CNN can attain high accuracy, although it may take longer to process.

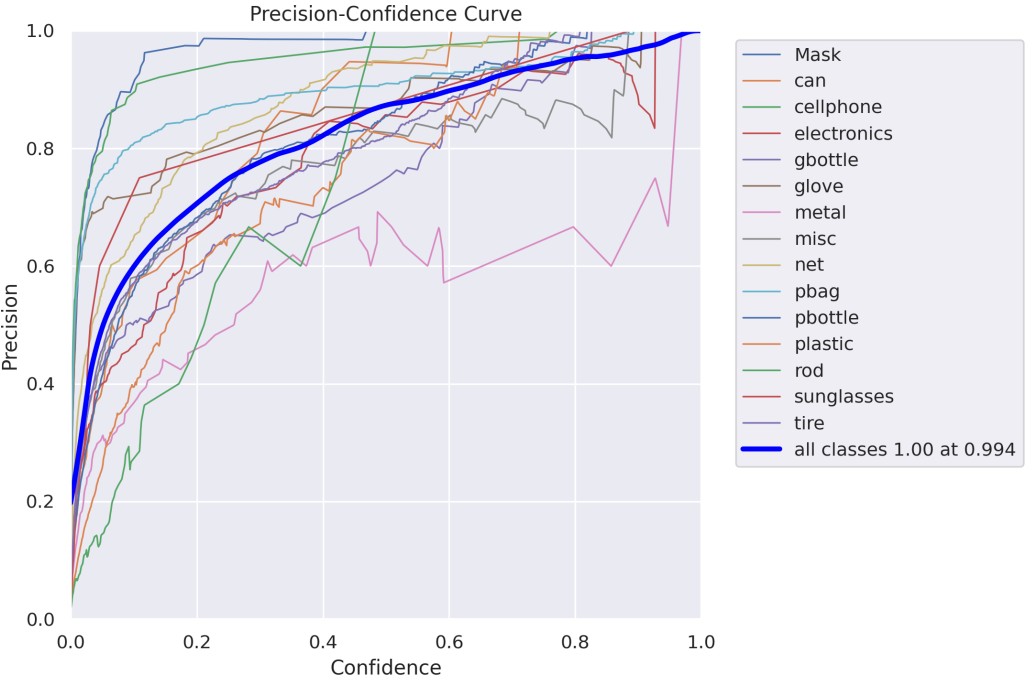

**Figure 8** Precision-confidence curve.

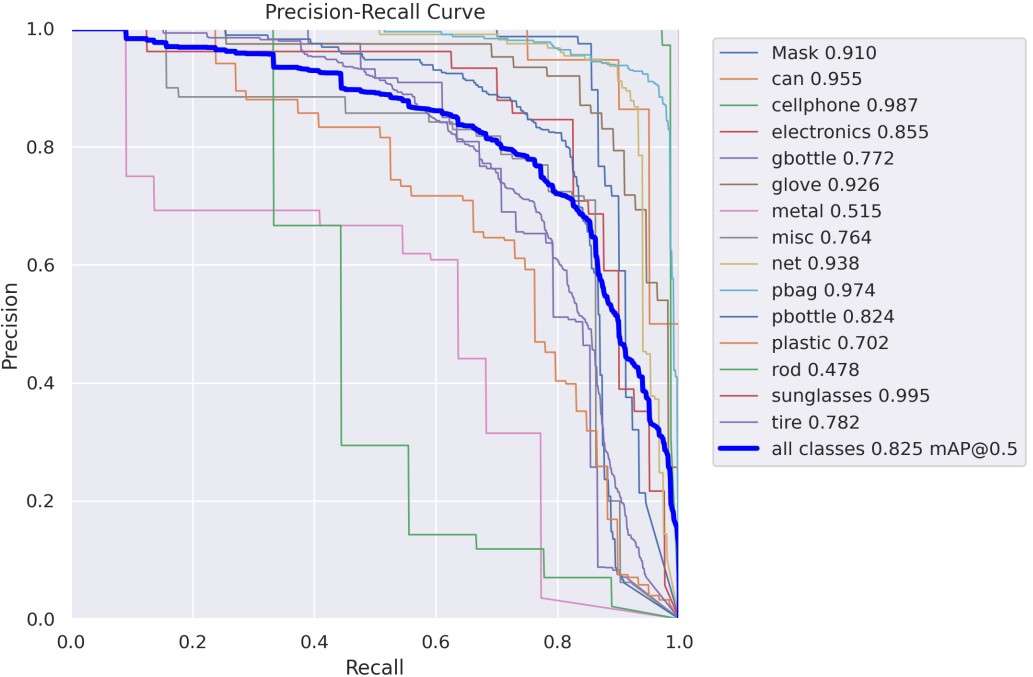

**Figure 9** Precision- recall curve.

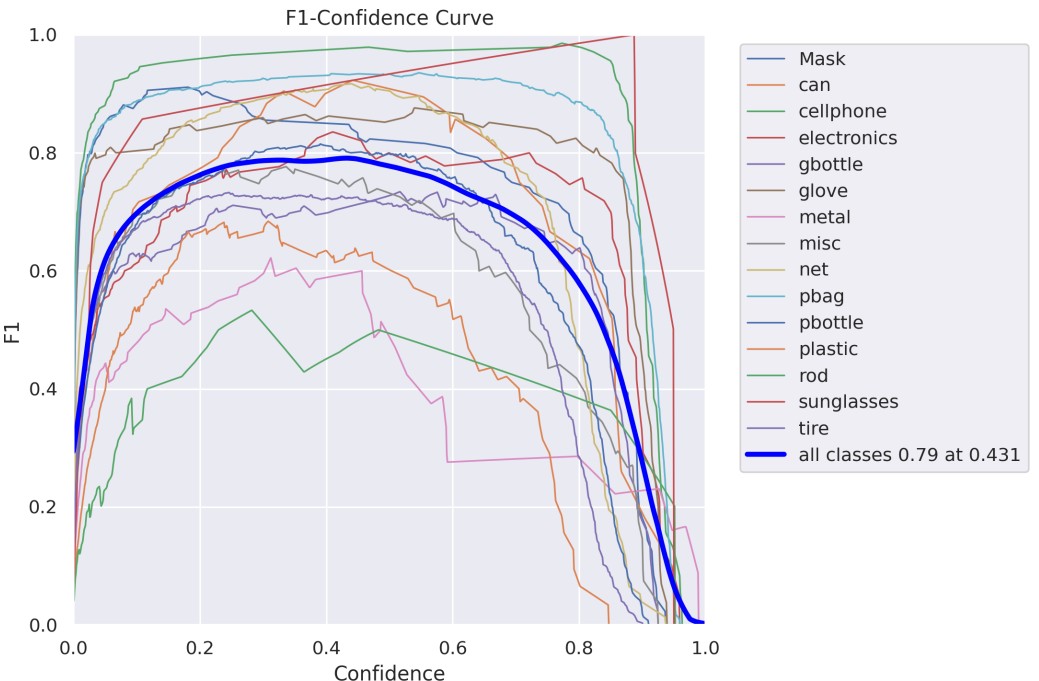

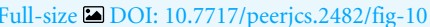

Figure 10    F1-confidence curve.

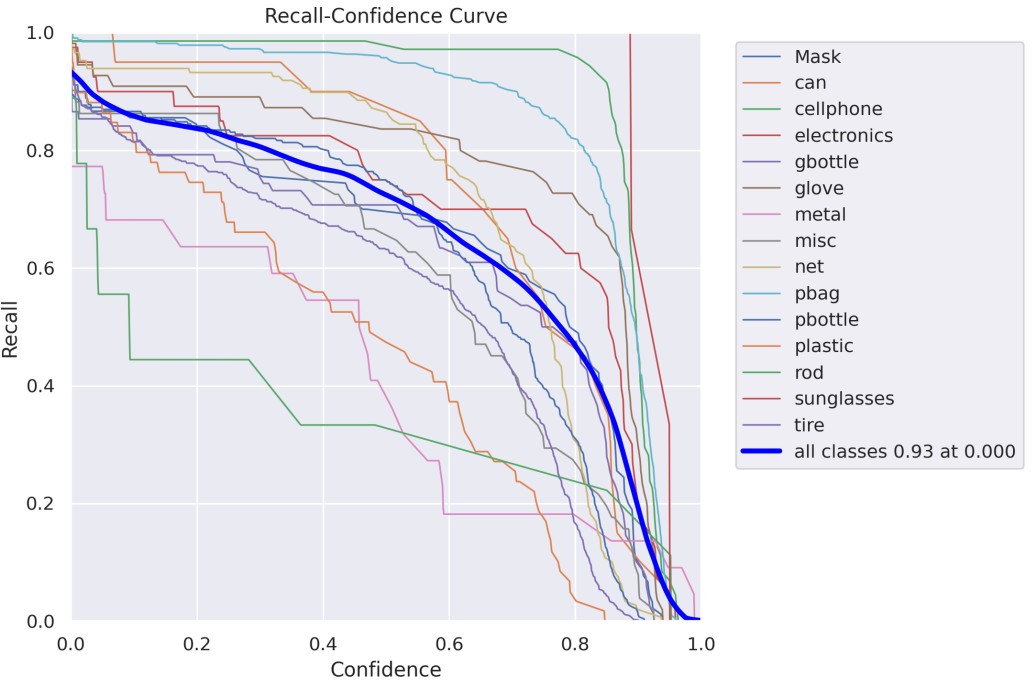

Figure 11    Recall-confidence curve.

**Table 7** **Ablation study.** Hyperparameter sensitivity analysis.

| S. No. | Experiments | Result | Impact |
|---|---|---|---|
| 1 | Base Model without optimized hyper-parameters tuning | 85.21% | – |
| 2 | Base Model with optimized Learning Rate | 88.16% | +2.95% |
| 3 | Base Model with optimized Batch Size | 89.79% | +1.63% |
| 4 | Base Model with optimized Momentum | 93.48% | +3.69% |
| 5 | Base Model with optimized IoU Threshold | 93.99% | +0.51% |
| 6 | Base Model with optimized Learning Rate & Batch Size | 94.87% | +0.88% |
| 7 | Base Model with optimized Learning Rate & Momentum | 95.01% | +0.14% |
| 8 | Base Model with optimized Learning Rate, Batch Size & Momentum | 95.85% | +0.84% |
| 9 | Base Model with optimized Learning Rate, Momentum & IoU Threshold | 97.05% | +1.20% |
| 10 | Optimized Model with optimized hyper-parameters tuning | 98.38% | +4.90% |

## Training and dataset requirements

Both the modified YOLO model and CNN require training, although for different datasets. A sizable labeled dataset including annotated bounding boxes of marine pollution items is required for the improved YOLO model. The algorithm is trained using this dataset in order to effectively detect and categorize pollution. During training, the YOLO model's object recognition and classification parameters are fine-tuned.

CNNs are often trained to classify images, rather than identify objects, during the training phase. For this, the users need a tagged set of marine pollution photographs, ideally with bounding box annotations. While CNN's ability to divide the world into various categories may be enough for some uses, it can't replace the pinpoint localisation information offered by object detection.

## Speed and efficiency

Faster and more accurate picture processing is a major benefit of the updated YOLO model. For applications where fast detection is critical, including real-time marine pollution monitoring, YOLO models are well-suited due to their ability to analyse photos in real-time or near real-time. On the other hand, when employed at a high resolution or with complicated designs, CNN may necessitate more computing and can be slower in analyzing pictures.

## Adaptability and generalization

The CNN is a flexible deep learning architecture that may be used for more than only object recognition in the realm of image analysis. It may be modified for use in evaluating marine contamination by segmenting images and extracting features. CNN models are well-suited for identifying subtle fluctuations and trends in marine pollution due to their capacity to learn complicated properties from photos.

The updated YOLO model may be limited in its ability to perform tasks other than localization and classification because of its focus on object identification. Image segmentation and in-depth study of pollution patterns are two examples of jobs for which

it might not be optimal. However, YOLO models have been employed well in a number of real-world applications, including the detection of marine pollution and other items.

Depending on the parameters of the marine pollution prediction job, either the modified YOLO model or CNN may be selected. While CNN is flexible and may be used for a variety of image processing applications, the modified YOLO model is capable of fast and accurate real-time object recognition. In order to make a well-informed decision, it is important to think about the specifics of the dataset, the accuracy required, the necessity for real-time processing, and the goals of the application.

The authors constructed a new model architecture by adapting and merging deep learning approaches for marine pollution data. A hybrid network architecture using CNNs and RNNs helps interpret marine pollution data temporal and spatial trends. This strategy uses advanced feature engineering to improve model prediction. We employ satellite photos and oceanographic data to introduce new concepts to this subject. Ensemble learning was utilised to make more accurate predictions. Aggregating predictions from numerous models improves system performance and reduces variation. Because marine pollution data is scarce and imbalanced, we created synthetic data samples using cutting-edge data augmentation technologies. By doing so, we can improve model generalisability and address small dataset issues. Our real-time marine pollution prediction system uses deep learning models and a stakeholder-friendly interface. The system is presented in our study. This technology enables proactive ocean management and faster decision-making. Through case studies in several maritime ecosystems, we have shown our model accurate. These case studies demonstrate our technique in real-world situations, helping us understand marine pollution dynamics.

## DISCUSSION

The suggested sensor system for training FCN is now being used to acquire additional real-time, actual data. An ocean basin containing exclusively floating plastic samples will be used for data collection in the near future, as will data collected from a pending research vessel excursion in southern North Sea waters. The study would add to the growing works (*Chen et al., 2021*) on powers of artificial intelligence in prediction of ocean polution and related events. With this model, it seems more helpful in tourist places. Such places are ruined by plastic trash see a drop in revenue as a result. There are also substantial financial expenditures associated with cleaning and maintaining the facilities. Plastic trash on beaches have a detrimental effect on the economy, wildlife, and human health.

Assuring the long-term viability and good health of our oceans hinges on our capacity to accurately anticipate marine pollution utilizing a proposed model. Our goal here is to go more deeply into the salient features and ramifications of this strategy. The flexibility of a hybrid artificial intelligence (AI) approach to marine pollution prediction to combine several AI methods is one of its main benefits. Historical data may be analyzed using machine learning methods like decision trees, random forests, and support vector machines to reveal trends that lead to pollution incidents. Convolutional neural networks, a type of deep learning model, can process large amounts of unstructured data, making

it possible to spot tiny pollution symptoms. The prediction models are improved by the incorporation of expert systems because they are more interpretable and transparent due to the incorporation of domain knowledge and expert input. The hybrid method makes it possible to analyze and understand data from a wide variety of sources by employing these AI methods. Satellite images, ocean sensors, weather patterns, data from pollution stations, and findings from citizen science projects are all examples. The accuracy and reliability of the prediction models can be improved by combining several data streams for a more comprehensive knowledge of maritime pollution.

The proposed model's real-time monitoring and prediction skills are essential for making smart decisions and reducing marine pollution in a timely manner. Stakeholders will be better able to respond quickly to pollution incidents, deploy resources for containment and cleaning, and implement preventative measures to lessen the impact on marine ecosystems if early warnings and alarms are provided. This preventative measure is critical for long-term ocean health because it lessens the impact of pollution and helps maintain marine biodiversity. The hybrid AI method may be used to forecast marine pollution in a variety of contexts. For instance, it may be put to use foreseeing oil spills and facilitating quick responses to mitigate the resulting ecological harm. Harmful algal blooms are a serious problem for marine life and ecosystems, but this method can help forecast when they will occur, allowing us more time to prepare. In addition, AI prediction models may help pinpoint plastic pollution hotspots, which can guide cleaning initiatives and policy changes aimed at reducing plastic waste's entry into the ocean.

While there is much to be gained by adopting a this proposed model, there are also obstacles that must be overcome. As it is sometimes difficult to gather accurate and complete data on marine pollution, availability and quality of data remain significant obstacles. There needs to be an increase in the quality and consistency of collected data, as well as an increase in the ease with which stakeholders may share and use that data. The public's trust and confidence in AI-driven prediction models may be enhanced by taking into account ethical factors like data privacy and security. Concerns about the interpretability and explainability of AI models also need to be addressed. In order to win over skeptical stakeholders and policymakers, it is essential to explain the logic behind AI's predictions, especially when using deep learning models. New strategies for improving the explainability of models, such as those for attributing features and extracting rules, can be used to deal with this problem. For the hybrid AI strategy to be widely adopted, researchers, policymakers, and industry stakeholders must work together. Sharing knowledge, information, and tools within a team helps everyone create more reliable forecasting models. The predictions made by the hybrid AI technique may also be used by politicians to make evidence-based decisions and shape rules targeted at lowering marine pollution.

The use of a proposed model for marine pollution prediction might make a substantial impact on long-term ocean health. This method integrates machine learning algorithms, deep learning models, and expert systems to analyze many data streams in real time and accurately anticipate when and where pollution will occur. The full promise of this strategy, however, cannot be realized unless issues of data availability, interpretability,

and cooperation are resolved. Protecting and conserving our seas for future generations requires ongoing study, technological improvements, and collaborative actions.

## CONCLUSION AND FUTURE SCOPE

The study presented a technique that could help reduce water pollution and indeed detection and removal of pollutants. The study proves the power of Artificial intelligence in marine garbage detection. The YOLOv5 is a successful method even with a small number of datasets, particularly underwater. The existing system can be scaled up because it improved testing accuracy. A hybrid AI approach to marine pollution prediction could maintain ocean health over time. Marine contamination is dynamic and complex, making typical data collection and analysis methods insufficient. By mixing machine learning techniques, deep learning models, and expert systems, hybrid AI may investigate several data sources and provide more accurate and reliable prediction models. Using hybrid AI to anticipate marine pollution has many benefits. It enhances real-time pollution monitoring and prediction, speeding response and mitigation. Identifying pollution data patterns and trends with machine learning algorithms allows early identification and prevention. Deep learning techniques illuminate pollution processes by analysing huge and complex data. Expert systems with subject knowledge and competence can help explain AI models that improve decision-making. Hybrid AI has showed promise in case studies and real-world applications. Real-time oil spill monitoring, harmful algal bloom predictions, and plastic pollution hotspot identification are successful uses. If pollution episodes can be accurately forecast, stakeholders can prevent pollution, protect marine ecosystems, and ensure the ocean's long-term health.

This proposed model has overcome challenges before it can be extensively adopted. Large, high-quality data sets are scarce, but accurate forecasts require them. Making data collection and exchange more efficient is crucial. Stakeholders and policymakers trust and accept this model based on interpretability and explainability. The hybrid model must be combined with decision support systems to make better, more informed decisions in the future. Researchers, governments, and industry stakeholders must collaborate to solve marine pollution's many issues. Policy repercussions can drive AI technology integration into environmental management systems. Predicting marine pollution with mixed AI is a huge step towards long-term ocean protection. AI can improve marine pollution awareness, enable quick responses, and protect marine ecosystems' fragile balance. To fully realise the hybrid AI strategy's promise, we need continued research, technical advances, and collaboration to protect our seas and future generations.Hundreds of marine species have been directly impacted by plastic trash due to ingestion, asphyxia, and entanglement. Seabirds, whales, fish, and turtles, among others, often choke to death after ingesting plastic trash that they mistook for food. Aside from being unable to swim and having internal injuries, they also receive lacerations and infections. Because of their buoyancy, floating plastics can also serve as a vector for the spread of invasive marine species, which poses a direct danger to marine ecosystems and biodiversity. In the future, we intend to collaborate with relevant authorities to adopt this solution to complement efforts geared towards

water protection and indeed preservation of our waters. In future, we will also develop an AI-driven app for promotion of ocean health awareness.

### Funding
This research is funded by the Research Supporting Project number (RSPD2024R553), King Saud University, Riyadh, Saudi Arabia. Study design. The funder did not have a role in the data collection and analysis, decision to publish, or preparation of the manuscript.

### Grant Disclosures
The following grant information was disclosed by the authors:
Research Supporting Project number: RSPD2024R553.
King Saud University, Riyadh, Saudi Arabia.

### Competing Interests
Khursheed Aurangzeb is an Academic Editor for PeerJ Computer Science. Kuldeep Kumar is an employee of Fourien Inc. Alberta, Edmonton, Canada. Bijeta Seth employee of Bhagwan Mahaveer Institute of Engineering & Technology, Sonipat, Haryana. Previously Kuldeep was an employee of Xebia. Bijeta Seth is working in the Dept. of Computer Science and Engineering, B.M. Institute of Engineering & Technology, Sonipat, Haryana, an academic institution. She is not associated with Xebia in past and present.

### Author Contributions
- Michael Onyema Edeh conceived and designed the experiments, analyzed the data, authored or reviewed drafts of the article, and approved the final draft.
- Surjeet Dalal conceived and designed the experiments, performed the experiments, performed the computation work, prepared figures and/or tables, and approved the final draft.
- Musaed Alhussein conceived and designed the experiments, performed the experiments, analyzed the data, performed the computation work, prepared figures and/or tables, authored or reviewed drafts of the article, and approved the final draft.
- Khursheed Aurangzeb conceived and designed the experiments, performed the experiments, analyzed the data, prepared figures and/or tables, authored or reviewed drafts of the article, and approved the final draft.
- Bijeta Seth conceived and designed the experiments, performed the experiments, analyzed the data, performed the computation work, prepared figures and/or tables, authored or reviewed drafts of the article, and approved the final draft.
- Kuldeep Kumar conceived and designed the experiments, performed the experiments, analyzed the data, performed the computation work, authored or reviewed drafts of the article, and approved the final draft.

### Data Availability
The Marine Pollution dataset is available at: https://tuvalu-data.sprep.org/dataset/marine-pollution-pacific-1.

## Supplemental Information

Supplemental information for this article can be found online at http://dx.doi.org/10.7717/peerj-cs.2482#supplemental-information.

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
