# Peer review of "A novel deep learning model for predicting marine pollution for sustainable ocean management"

_PeerJ Computer Science, doi:10.7717/peerj-cs.2482_

## Round 0.1 · original submission · Major Revisions

All the reviewers raised concerns and even criticisms. As a result, the paper cannot be accepted in the present form. The editor would like to give the authors a chance to revise the paper and improve the quality of the paper.

Reviewer 1 ·

Basic reporting

The authors addressed marine plastic pollution by leveraging artificial intelligence to identify and categorize underwater plastic waste. They used Two-step Clustering and Fully Convolutional Network (FCN) algorithms, trained with Kaggle's in-situ plastic location data. There are some suggestions for the authors to improve their paper:

1. The summary of related research on marine pollution prediction is insufficient.

2. The models used by the authors are all well-established existing models and they do not focus on the contributions and innovations of their research.

3. The related work section occupies a significant amount of space and should be more concise and comprehensive rather than listing each study individually.

4. There are not enough figures, algorithms, and formulas to adequately introduce the model.

Experimental design

1. The experimental section lacks comparisons with existing state-of-the-art models and ablation studies.

Validity of the findings

no comment

Additional comments

1. Figure 3 is not clear enough.

Reviewer 2 ·

Basic reporting

The overall workload of the article is full and the experiments are sufficient, but there are the following minor problems:
(1) The pseudo-code of the YOLO algorithm is rather simple, it is recommended to write it in a fuller way to help the clarity of the article.
(2) The flowchart in the article is rather fuzzy, please give a high-definition picture.
(3) The references in the article do not have articles from 2022-2024, please update your references.

Experimental design

no comment

Validity of the findings

no comment

Additional comments

no comment

·

Basic reporting

1. References for the manuscript are not selected properly. Maximum references are from year 2020 and none of the reference is from year 2023 and year 2024. This does not show good selection and distribution of the references. As the references from the recent years are not covered, we cannot say that this is the most recent, innovative and needed work carried out by the authors.
2. Lots of references mentioned under the list of references are not cited in the manuscript and that percentage is almost 50%. Each and every reference listed under the list of reference must be cited in the manuscript. Kindly look into it and make the required corrections.
3. In Figure No 8: What is X-axis and Y-axis title. Need to provide unit (if any) for both the X and Y axis titles. Remove the 'tag' from the figure as it is just repeating the above line.
4. In Figure No 9: What is X-axis and Y-axis title. Need to provide unit (if any) for both the X and Y axis titles. Remove the 'tag' from the figure as it is just repeating the above line.
5. In Figure No 10: What is X-axis and Y-axis title. Need to provide unit (if any) for both the X and Y axis titles. Remove the 'tag' from the figure as it is just repeating the above line.
6. Lots of formatting related changes are required throughout the manuscript.
7. At lots of places the acronyms provided are not having the correct longform associated with it. Lot of repetition of the mention of longform of acronyms whenever they appear (e.g CNN, AI, FCN etc), each time authors are mentioning its long form, that's not needed each time. Authors need to look into it.
8. Try to use a word 'Costal' instead of 'Coast' in the manuscript.
9. YOLO work must be written in capital letters being acronym whenever it appears in the manuscript. Kindly correct it wherever required.

Experimental design

1. How can be a percentage value equal to 983.33? Please check the values mentioned in the manuscript for the correctness.
2. In Figure 2: why is the aquatic life getting detected as a pollutant as per your set standards (0 to 2, 3 levels). Where as plastic bag is getting detected as '0' more clear as per the scale of 3 mentioned on line 232. Some explanation of the notations on the figure are not matching properly, as '0' being clear and '2' being pollutant as per your explanation. Kindly look into it and correct the same. Aquatic life should not be coming under the ocean pollution, which is currently coming in your algorithm with level '1'.
3. By looking at the results it feels currently your developed algorithm i doing just object detection, as it is able to exclude aquatic life form it.
4. No need to explain 'Data Collection and Preprocessing' two times for two different methods. Because that base almost remains same irrespective of the method used. So kindly remove the repetition of lines 355 to 361.
5. Equation No 4: The formula for the 'Recall' is not correct. Kindly correct it.

Validity of the findings

1. The claims made by the authors in the conclusion part are not justifiable.
2. The technique/algorithm developed by the authors is not helping in reducing the water pollution, it is just detecting the pollutants upto certain extent in the sea water. Please reform the statements used in the conclusion section.
3. No procedure to remove the pollutants detected by the system is explained in the manuscript. So authors cannot claim the technology is providing reduction in water pollution and removal pollutants. Your technique is currently only detecting the pollutants in the sea water upto certain extent, this is the actual scope of your work carried out.

Additional comments

1. References part requires big time updation.
2. Results must be correct and should not be having any ambiguity.
3. Conclusion should be as per the work done and the results obtained, not beyond that.

---

## Round 0.2 · Major Revisions

Two reviewers are happy with the paper but one reviewer gave negative recommendations. As a result, the paper needs a major revision.

Reviewer 1 ·

Basic reporting

1. In the related work section, the authors spend too much text on descriptions. They should categorize the existing research and provide a concise yet comprehensive summary of the studies.

Experimental design

1. The results in Table 7 cannot be considered an ablation study, as parameters like learning rate and batch size are not part of the model itself. The setting of these hyperparameters should be determined through sensitivity analysis.

Validity of the findings

no comment

Additional comments

1. The contributions of this paper are relatively weak, and it is not clearly explained in the paper how the authors modified YOLO.

2. There is a lack of sufficient theoretical support; the authors did not provide any formulas or theoretical explanations for either the two-step clustering or the YOLO model.

Reviewer 2 ·

Basic reporting

no comment

Experimental design

no comment

Validity of the findings

no comment

Additional comments

Thank you for addressing all the comments provided with details. Authors have addressed all the review comments satisfactorily.

·

Basic reporting

Now the required suggestions are reflected by the authors at the appropriate place.

Experimental design

Corrections suggested in the formula are now taken care of by the authors.

Validity of the findings

Ok

Additional comments

no

---

## Round 0.3 · accepted · Accept

Congratulations! Your paper has been accepted for publication.